# miR155 Deficiency Reduces Myofibroblast Density but Fails to Improve Cardiac Function after Myocardial Infarction in Dyslipidemic Mouse Model

**DOI:** 10.3390/ijms22115480

**Published:** 2021-05-22

**Authors:** David Schumacher, Adelina Curaj, Sakine Simsekyilmaz, Andreas Schober, Elisa A. Liehn, Sebastian F. Mause

**Affiliations:** 1Institute for Molecular Cardiovascular Research (IMCAR), University Hospital Aachen, RWTH Aachen University, 52074 Aachen, Germany; dschumacher@ukaachen.de (D.S.); acuraj@ukaachen.de (A.C.); sakine@gmx.de (S.S.); 2Department of Anesthesiology, University Hospital Aachen, RWTH Aachen University, 52074 Aachen, Germany; 3Victor Babes National Institute of Pathology, 050096 Bucharest, Romania; 4Institute for Prevention and Epidemiology of Cardiovascular Disease, University of Munich, 80539 München, Germany; Andreas.Schober@med.uni-muenchen.de; 5Department of Cardiology, Pulmonology, Angiology and Intensive Care, University Hospital, RWTH Aachen University, 52074 Aachen, Germany; 6Human Genetic Laboratory, University for Medicine and Pharmacy, 200642 Craiova, Romania

**Keywords:** microRNA, cardiovascular disease, myocardial infarction, heart failure, dyslipidemia

## Abstract

Myocardial infarction remains the most common cause of heart failure with adverse remodeling. MicroRNA (miR)155 is upregulated following myocardial infarction and represents a relevant regulatory factor for cardiac remodeling by engagement in cardiac inflammation, fibrosis and cardiomyocyte hypertrophy. Here, we investigated the role of miR155 in cardiac remodeling and dysfunction following myocardial infarction in a dyslipidemic mouse model. Myocardial infarction was induced in dyslipidemic apolipoprotein E-deficient (ApoE^−/−^) mice with and without additional miR155 knockout by ligation of the LAD. Four weeks later, echocardiography was performed to assess left ventricular (LV) dimensions and function, and mice were subsequently sacrificed for histological analysis. Echocardiography revealed no difference in LV ejection fractions, LV mass and LV volumes between ApoE^−/−^ and ApoE^−/−^/miR155^−/−^ mice. Histology confirmed comparable infarction size and unaltered neoangiogenesis in the myocardial scar. Notably, myofibroblast density was significantly decreased in ApoE^−/−^/miR155^−/−^ mice compared to the control, but no difference was observed for total collagen deposition. Our findings reveal that genetic depletion of miR155 in a dyslipidemic mouse model of myocardial infarction does not reduce infarction size and consecutive heart failure but does decrease myofibroblast density in the post-ischemic scar.

## 1. Introduction

Myocardial infarction (MI) with subsequent cardiac remodeling represents a major source of morbidity and mortality and remains the most common cause of heart failure. Addressing adverse remodeling has, therefore, become an attractive but still challenging therapeutic option. As recent studies suggest a critical involvement of microRNA (miR) in healing and repair processes after myocardial injury, targeting of miR signal pathways appears to be a promising therapeutic strategy for beneficially influencing cardiac remodeling [1,2].

Non-coding miR155 has been identified as a relevant regulatory factor in vascular diseases and cardiac remodeling. As such, miR155 expression in macrophages may promote vascular inflammation and affect atherosclerosis [3,4,5]. ApoE^−/−^ mice with a leukocyte-specific miR155 deficiency, additional miR155 knockout or treated with antagomir-155 displayed a decreased plaque size and lower number of lesional macrophages. However, dyslipidemic low density lipoprotein receptor deficient (Ldlr^−/−)^ mice transplanted with miR155-deficient bone marrow displayed increased atherosclerotic plaques [5]. Confirming stage- and context-dependent effects on atherogenesis and inflammation, it was shown that miR155 may suppress macrophage proliferation and protect against early atherosclerosis, but also that it promotes advanced atherosclerosis [6]. 

In mouse models with stressor-induced hypertrophy, a genetic deficiency or pharmacological inhibition of miR155 attenuated inflammation, fibrosis and hypertrophy of the heart and alleviated cardiac dysfunction [7,8]. Furthermore, miR155 may context-dependently influence apoptosis and cellular differentiation and may thus modulate tissue remodeling [9]. In normolipidemic mice, treatment with miR155 inhibitors resulted in a less pronounced impairment of left ventricular function just 1 or 3 days after induction of myocardial infarction [10,11]. However, long-term effects with completed myocardial scar formation have not been investigated thus far. Interestingly, elevated blood levels of miR155 at early time points following acute MI correlated with adverse remodeling and increased mortality [12,13]. Furthermore, serum levels of miR155 positively correlate with post-MI heart failure, suggesting that miR155 might represent a potential prognostic marker following MI [14]. Of note, several studies on miR155 documented ambivalent and partly contradictory effects on various aspects of vascular and cardiac diseases, regenerative processes and cellular behavior such as apoptosis, viability and differentiation [4,15]. This may reflect the fact that the role of miR155 in disease initiation and progression is highly dependent on the pathological context, due to distinct disease- and cell-specific signal pathways as well as distinct temporospatial regulation of signal transduction and the stage of the pathology. 

As dyslipidemia is often present in patients with myocardial infarction and is known to have detrimental effects on remodeling and postinfarction cardiac function [16,17], using ApoE^−/−^ mice, we investigated the effect of miR155 deficiency on cardiac function and remodeling following myocardial infarction in the context of dyslipidemia.

## 2. Results

### 2.1. miR155 Knockout Does not Improve Heart Function after Myocardial Infarction in ApoE^−^/^−^ Mice

To investigate the role of miR155 in healing and functional recovery after myocardial infarction under dyslipidemic conditions, we induced myocardial infarction by permanent ligation of the LAD in ApoE^−/−^ and ApoE^−/−^/miR155^−/−^ mice fed a normal chow diet. Together with the observation that the heart weight/body weight ratio of ApoE^−/−^/miR155^−/−^ mice did not differ from that of the ApoE^−/−^ controls (5.2 ± 0.7 vs. 5.0 ± 0.9 mg/g), this demonstrates that a genetic deficiency of miR155 does not affect heart development and cardiac function in dyslipidemic mice. Furthermore, in a previous study, we showed that serum cholesterol levels were not affected by additional miR155 deficiency in dyslipidemic ApoE^−/−^ mice [6].

Four weeks after LAD occlusion, infarction size, as determined by Gomori trichrome staining, was not different between ApoE^−/−^ mice with and without additional miR155 knockout (Figure 1A,B). Echocardiography performed just before sacrifice confirmed the impairment of cardiac function compared to the baseline, but revealed no difference in LV ejection fraction or end-diastolic and end-systolic LV volume (Figure 1C). LV wall thickness and the calculated LV mass were comparable between both groups (data not shown). In addition, the heart rate of the mice was independent of miR155 deletion at unaltered levels. Altogether, these data show that miR155 deletion does not protect from cardiac deterioration and does not alter hemodynamics following the induction of myocardial infarction in dyslipidemic mice.

### 2.2. miR155 Knockout Decreases Myofibroblast Density in the Scar but Does not Influence Collagen Amount and Neoangiogenesis

In order to assess postinfarction healing and fibrosis in ApoE^−/−^ mice, we performed a histopathologic analysis of paraffin-embedded sections of the heart following Gömöri trichrome and immunofluorescence staining. Four weeks after myocardial infarction, the amount of smooth muscle actin (SMA)-positive myofibroblasts was significantly decreased in the infarcted area of ApoE^−/−^/miR155^−/−^ mice compared to that of ApoE^−/−^ mice (Figure 2A). However, miR155 knockout had no effect on general collagen deposition and fibrosis (Figure 2B). To assess postinfarction neoangiogenesis, we performed CD31 immunofluorescence staining for the detection of endothelial cells. Of note, it was found that miR155 deletion had no effect on the density of capillaries in the infarcted area (Figure 2C). These data reveal that loss of miR155 modifies the composition of the infarcted tissue to a lower density of myofibroblasts, but does not alter levels of postinfarction fibrosis and angiogenesis. 

## 3. Discussion

Despite substantial advances in medical and device therapy, heart failure following myocardial infarction and subsequent cardiac remodeling remains a challenging clinical problem linked to morbidity and mortality. miR155 is upregulated following myocardial infarction and may critically regulate processes of cardiac remodeling [11,18]. In mouse models, miR155 was shown to promote cardiovascular cardiac inflammation, stage-dependently affect atherosclerosis and potentially induce pathological hypertrophy with concomitant heart failure [3,4,7]. Here, we investigated, in the context of dyslipidemia, the role of miR155 in cardiac remodeling and development of ischemic heart failure following ligation-induced myocardial infarction. Our data revealed that miR155 deletion in a dyslipidemic mouse model affects the composition of the infarcted tissue and suggest that miR155 is involved in the regulation of myofibroblasts in the infarcted area, but fails to confer a beneficial effect on postinfarction cardiac dysfunction and is unable to modify cardiac fibrosis and angiogenesis in the scar. 

Previous animal studies in a normolipidemic context suggested a protective role of transient miR155 inhibition in early cardiac dysfunction and remodeling shortly after myocardial infarction [10,11]. Several conceptual and methodological differences might contribute to the discrepant findings. In contrast to our study, antagomir-155 was used for inhibition of miR155 and was applied shortly before the induction of myocardial infarction. Transient inhibition using anti-miR chemistries may differ from miR knockout models, as genetic deletion of a specific miR throughout embryonic and postnatal development allows for compensatory mechanisms that substitute for that function [19]. Notably, such mechanisms are not engaged under conditions of short-term inhibition with antagomirs. Furthermore, cardiac function and infarction size were analyzed as early as one to three days following ligation of the LAD. At such early stages, remodeling of the hypoxic myocardium is far from complete, and the respective tissue still undergoes multiple compositional and structural changes, each governed by stage-specific and context-specific processes and signaling. Lastly, previous experiments with antagomir-155 were performed in a normolipidemic context. Independent of additional miR155 knockout, our ApoE^−/−^ mice fed a normal chow diet were characterized by substantial hypercholesterolemia and dyslipidemia. Dyslipidemia is a major risk factor for atherosclerosis, and subsequent acute cardiovascular events are a common feature in patients with myocardial infarction. Additionally, dyslipidemia per se is known to have detrimental effects on postinfarction cardiac function [16,17]. Functional analysis of miR155 deletion in an infarction model with dyslipidemia might more adequately reflect real-world scenarios and may, therefore, be a more suitable approach to derive conclusions of clinical relevance. Of note, a recent study detected a negative correlation between the concentration of circulating miR155 and cholesterol levels in humans [20]. With such a scenario, hyperlipidemia-associated reduction in miR155 expression might diminish, in our study, the effects of genetic miR155 deletion on cardiac remodeling. 

Analysis of the postinfarction scar revealed a lower density of SMA-positive myofibroblasts in mice with an miR155 deficiency. This is in line with previous findings demonstrating that depletion of miR155 downregulates the expression of SMA in the myocardium and inhibits fibroblast proliferation and differentiation into myofibroblasts [21]. Emphasizing a role in fibrosis, miR155 depletion has been shown to abrogate adverse cardiac remodeling in various mouse models and suppress cardiac hypertrophy in response to various pathological stressors [7,8]. Myofibroblasts as homeostatic regulators of the extracellular matrix are integrally involved in various aspects of the repair and remodeling of the heart that occurs following MI [22] and are essential for maintaining its structural integrity. In temporospatial dependence, myofibroblasts may confer both beneficial and detrimental cardiac effects [23]. Manifesting their ambivalent role, a high density of myofibroblasts may contribute to the formation of a robust and stable scar; however, excessive myofibroblast activation and persistence may drive fibrosis and myocardial stiffness, potentially resulting in heart failure progression. 

It is known that during advanced maturation of a postinfarction scar, the density of myofibroblasts significantly decreased due to accelerated apoptotic cell death [24]. The presently observed impaired persistence of myofibroblasts in miR155-deficient mice leads to the speculation that genetic deletion of miR155 may accelerate apoptosis and attenuate the viability of myofibroblasts. Indeed, inhibition of miR155 has generally been associated with pro-apoptotic signaling and may suppress the ability of cells to differentiate [25,26]. Conversely, overexpression of miR155 was shown to attenuate necrotic cell death of cardiomyocyte progenitor cells and to promote proliferation and survival of colon cancer cells [27,28]. However, data regarding the role of miR155 in apoptosis and cell cycle progression are currently still conflicting, suggesting a possible context-dependent and cell-specific effect of miR155 and indicating the need for further studies [11,29]. The unaltered collagen content in the myocardial scar as well as the unchanged LV mass and relative heart weight confirm ongoing and persistent secretory activity in miR155-deficient mice and may indicate a compensatory augmentation of collagen synthesis by myofibroblasts. Of note, the striking uncoupling of myofibroblast density and collagen content found in miR155-deficient mice may alternatively reflect impaired maturation and differentiation of secretory active cardiac fibroblasts to SMA-positive myofibroblasts.

The role of miR155 in angiogenesis and neovascularization remains unclear, as both pro- and anti-angiogenic effects have been attributed to miR155 in different scenarios such as acute ischemia and neoplasia [30,31]. In our study, we observed that miR155 deletion had no effect on adaptive neovascularization, with an unaltered density of capillaries in the postischemic myocardium.

In summary, our study provides evidence that in the context of dyslipidemia, genetic depletion of miR155 affects the cellular composition of the matured infarcted myocardium with decreased density of myofibroblasts but does not alter postinfarction cardiac function or fibrosis and angiogenesis in the scar. To further unravel the regulatory effect of miR155 in postinfarction remodeling and cardiac regeneration, continuing studies are needed to identify cell-specific miR155 expression patterns and explore the complex temporal and spatial aspects of miR155 involvement in phenotype control and viability of myofibroblasts, macrophage-governed myocardial inflammation and promotion of angiogenesis. Such assessment, together with investigation of the microenvironmental determinants that critically influence the role of miR155, may help to better define the functional consequences of miR engagement in postischemic myocardial remodeling under various conditions such as dyslipidemia and other metabolic disorders.

## 4. Materials and Method

All animal experiments were performed in accordance with European legislation and approved by local German authorities (LANUV—Landesamt für Natur, Umwelt und Verbraucherschutz Nordrhein-Westfalen, approval number: AZ:84-02.04.2013.A185, approval date: 16 August 2013). All mice were housed under standardized conditions in the Animal Facility of the University Hospital Aachen (Aachen, Germany).

### 4.1. Mouse Model of Myocardial Infarction

Eight- to ten-week-old male C57Bl/6 Apolipoprotein E knockout (ApoE^−/−^) and C57Bl/6 double ApoE and miR155 knockout (ApoE^−/−^/miR155^−/−^) mice fed a chow diet underwent myocardial infarction as previously described [32,33,34,35]. In brief, mice were intubated under general anesthesia (100 mg/kg ketamine, 10 mg/kg xylazine, i.p.) and analgesia (0.1 mg/kg Buprenorphine) and ventilated using positive pressure and oxygen using a rodent respirator. The heart was exposed by left side thoracotomy, and myocardial infarction (MI) was induced by permanent ligation of the proximal left anterior descending artery (LAD). The rib, muscle and skin incisions were closed with separate sutures. Analgesia was continued for another 5 days after MI using 0.1 mg/kg of Buprenorphine. 

### 4.2. Echocardiography

Before and four weeks after MI, left ventricular (LV) function and LV volumes were determined by echocardiography performed on a small-animal ultrasound imager (Vevo 770, FUJIFILM Visualsonics, Toronto, Canada). During the procedure, mice were sedated with 1.5% isoflurane. Measurements of the short and long parasternal axes were taken in B-Mode (2D-realtime) and M-Mode using a 40-megahertz transducer. The LV ejection fraction (EF) was assessed and analyzed in the long axis and in the short axis, obtained at the level of the papillary muscles. Furthermore, we recorded and calculated fractional shortening (FS), systolic and diastolic LV volumes, LV wall thickness, LV mass and heart rate (HR).

### 4.3. Histology and Immunohistochemistry 

For histological analysis of the MI, mice were sacrificed 4 weeks after ligation of the LAD. Animals were anesthetized (100 mg/kg ketamine, 10 mg/kg xylazine, i.p.) and the hearts were excised, fixed in formalin and embedded in paraffin. Serial sections (10–12 sections per mouse, 400 µm apart, up to the mitral valve) were stained with Gömöri trichrome (Abcam, Cambridge, UK, ab150686). The area of the infarcted tissue was assessed for all sections using Diskus software (Hilgers, Königswinter, Germany) and expressed as a percentage of total LV volume. Blue-stained collagen content was analyzed with Cell P Software (Olympus, Hamburg, Germany) and expressed as percentage of the infarcted area. Serial sections (3 sections per mouse, 400 µm apart) were stained to analyze the infarcted area for smooth muscle actin-positive myofibroblasts (SMA, DAKO/Agilent, Waldbronn, Germany) and CD31-positive capillaries (CD31, Santa Cruz, Dallas, TX, USA). Positive stained cells or vessels were counted in 6 different fields per section and expressed as cells or vessels per mm^2^. 

### 4.4. Statistical Analysis

Data representing mean ± SEM were analyzed using either Student’s *t*-test followed by a Tukey post hoc test (for normally distributed data) or by a nonparametric Mann–Whitney test followed by a post hoc Dunn test (for non-normally distributed data) using GraphPad Prism v6.1 for Windows (GraphPad) as appropriate. *p*-values of <0.05 were considered significant.

## Figures and Tables

**Figure 1 ijms-22-05480-f001:**
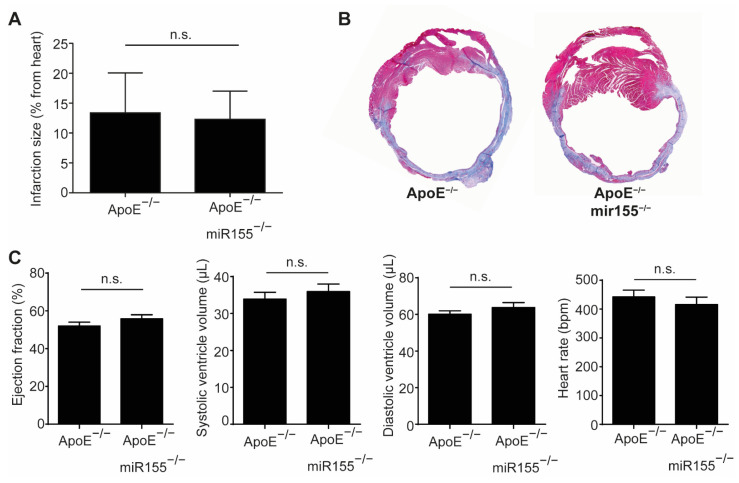
miR155 knockout does not improve cardiac function after myocardial infarction in dyslipidemic mice. (**A**) Myocardial infarction size in ApoE^−/−^ and ApoE^−/−^/miR155^−/−^ mice 4 weeks after ligation of the LAD and expressed as % of total heart. (**B**) Representative images of Gomori trichrome staining revealing the scar (blue) and healthy myocardium (red). (**C**) Echocardiographic analysis of LV ejection fraction, end-systolic and end-diastolic LV volume and heart rate 4 weeks after myocardial infarction. *n* = 8 for each condition. *p* > 0.05 was defined as n.s. (not significant).

**Figure 2 ijms-22-05480-f002:**
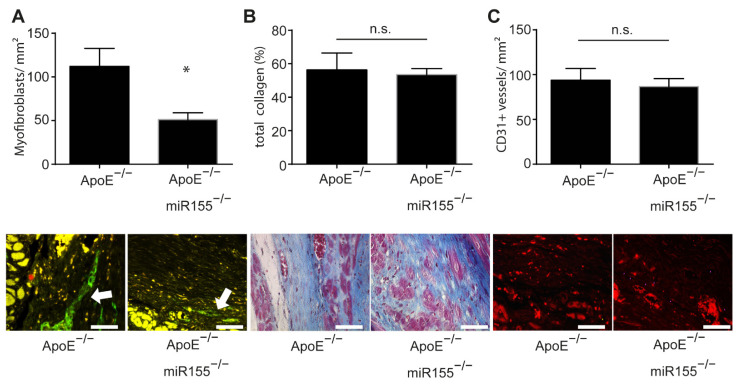
miR155 knockout decreases myofibroblast density in the scar but does not influence collagen amount and neoangiogenesis. (**A**) Amount of myofibroblasts per mm^2^ in the scar 4 weeks after myocardial infarction. Representative images of SMA immunofluorescence staining. * *p* < 0.05 vs. ApoE^−/−^ mice. (**B**) Total collagen content in the scar 4 weeks after myocardial infarction. Images of Gömöri trichrome staining of representative heart sections indicating myocardial collagen content (blue). (**C**) Amount of CD31-positive blood vessels per mm^2^ in the scar four weeks after myocardial infarction. Representative images of CD31 immunofluorescence staining (red). *n* = 8 for each condition. Scale bar: 50 μm. * *p*-values of >0.05 were defined as significant. *p*-values of <0.05 were defined as not significant (n.s.).

## Data Availability

The data presented in this study are available on request from the corresponding authors.

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
