# Peer review of "miR155 Deficiency Reduces Myofibroblast Density but Fails to Improve Cardiac Function after Myocardial Infarction in Dyslipidemic Mouse Model"

_ijms, 2021, doi:10.3390/ijms22115480_

Round 1
Reviewer 1 Report
Here, authors investigated role of MiR155 in cardiac remodeling and dysfunction following myocardial infarction under dyslipidemic condition. Considering growing interest of microRNA in myocardial infarction, this study is very timely and important. However, it is very imperative to improve this manuscript by addressing following points,
- in the introduction, its good to expand the literature describing the finding of others in mouse and human.
- Please mentioned the duration of diet before induction of LAD
- Since previous study show improvement in the myocardial infarction using miR155 antagomir in normolipidemic mice. It is interesting to see if knockout mice shows any effect under chow fed condition.
- What is the lipid levels before and after LAD in mice?
Author Response
Here, authors investigated role of MiR155 in cardiac remodeling and dysfunction following myocardial infarction under dyslipidemic condition. Considering growing interest of microRNA in myocardial infarction, this study is very timely and important. However, it is very imperative to improve this manuscript by addressing following points.
-> We sincerely thank the Referee for considering our study very timely and for providing constructive comments and suggestions, which helped us to substantially improve our manuscript.
Comment 1: In the introduction, its good to expand the literature describing the finding of others in mouse and human.
Response 1: Now we have expanded the literature in the Introduction Section to include relevant findings regarding the role of miR155 in current experimental mouse settings and various human studies, as suggested (line 53-59 and 67-71).
Comment 2: Please mentioned the duration of diet before induction of LAD
Response 2: We apologize for this mistake, the mice were fed a normal chow diet, not a western type of diet. Now we have corrected and clarified this issue in the Method (line 233), Results (line 87) and Discussion Section (line 165-167).
ApoE−/− mice fed a normal chow diet are known to develop substantial hypercholesterolemia with plasma total cholesterol levels around 13 mmol/L (500 mg/dL) with strikingly elevated VLDL (Getz GS et al., ATVB 2006; Oppi S et al., Front Cardiovasc Med. 2019). Of note, such ApoE−/− mice spontaneously develop quite complex and widespread atherosclerosis while being fed normal mouse chow diet.
Comment 3: Since previous study show improvement in the myocardial infarction using miR155 antagomir in normolipidemic mice. It is interesting to see if knockout mice shows any effect under chow fed condition.
Response 3: As mentioned before, the results presented in our study are under normal chow diet. Now we have corrected this issue in the Results (line 93-95) and Discussion Section (line 165-167). The results presented in our study reflect outcomes in ApoE−/− and ApoE−/−/miR155−/− mice fed a normal chow diet with substantial hypercholesterolemia.
Comment 4: What is the lipid levels before and after LAD in mice?
Response 4: We agree with the Referee that consideration of the lipid levels is relevant for the interpretation of the data. We have previously shown that serum cholesterol levels were not significantly affected by miR155 deficiency in ApoE−/− mice (Wei Y, Schober A et al, ATVB 2015; cholesterol levels about 18 mmol/L (700 mg/dL)). This is in accordance with the report by Donners M et al. (PLOS One 2012), demonstrating that no differences were found in either plasma cholesterol or triglyceride levels before and after high cholesterol diet feeding. This aspect is now commented in the Results (line 93-95) and explicitly outlined Discussion Section (line 165-167). Induction of MI does not relevantly modify lipid levels in the chronic phase.
Reviewer 2 Report
Authors found that genetic depletion of miR155 in ApoE-/- mice decreased the myofibroblast density 4 weeks after ligation of the LAD, but failed to reduce MI size and alter cardiac remodeling. Due to such preliminary and short data, it seems that authors submitted this work as a Communication article.
As authors stated in their article, compared to previous studies, they specifically performed this preliminary experiment using dyslipidemia model mice and at chronic state of MI. This reviewer wonders if genetic depletion of miR155 could alter any cardiac function and/or remodeling 4 weeks after occurrence of experimental MI in the normolipidemic mice as another control.
Whether the burdens of apoptosis and/or its relevant signaling were actually accelerated in the miR155 deficient mice could also strengthen your speculation.
Author Response
Authors found that genetic depletion of miR155 in ApoE-/- mice decreased the myofibroblast density 4 weeks after ligation of the LAD, but failed to reduce MI size and alter cardiac remodeling. Due to such preliminary and short data, it seems that authors submitted this work as a Communication article.
-> We thank the Referee for carefully examining our manuscript and are grateful for providing constructive and helpful comments, thus helping to improve the manuscript. We indeed intentionally submitted the manuscript as a Communication article due to the focused approach of our study.
Comment 1: As authors stated in their article, compared to previous studies, they specifically performed this preliminary experiment using dyslipidemia model mice and at chronic state of MI. This reviewer wonders if genetic depletion of miR155 could alter any cardiac function and/or remodeling 4 weeks after occurrence of experimental MI in the normolipidemic mice as another control.
Response 1: Epidemiological studies demonstrated that roughly 50% of the adult population has dyslipidemia. As outlined in the Discussion Section, dyslipidemia is a major risk factor for atherosclerosis and subsequent acute cardiovascular events such as myocardial infarction and is therefore a common feature in patients with myocardial infarction. In addition, dyslipidemia may contribute to heart failure following myocardial infarction. We therefore chose to investigate effects of genetic miR155 deficiency on post-myocardial remodeling in the context of dyslipidemia using ApoE−/− mice.
We now rephrased and expanded the respective statement in the Discussion Section (line 171) to emphasize the relevance of dyslipidemia for the evaluation of miR155 deficiency in remodeling and heart function following induction of myocardial infarction.
Comment 2: Whether the burdens of apoptosis and/or its relevant signaling were actually accelerated in the miR155 deficient mice could also strengthen your speculation.
Response 2: We completely agree with the Referee that apoptosis and its signaling are processes of substantial relevance for cardiac remodeling and development of the scar. Unfortunately, we didn’t perform cardiac tissue analysis at earlier time points following MI to adequately address assessment of apoptosis, since we didn’t observe any differences at the final end-time point 4 weeks after ligation of the LAD. However, the documented impaired persistence of myofibroblasts in miR155 deficient mice leads to the speculation that genetic deletion of miR155 may accelerate apoptosis and attenuate viability of myofibroblasts.
In respect to the comments of the Referee, we now more extensively comment this issue in the Discussion Section (line 198-201) citing the current known literature and acknowledging that data regarding the role of miR155 in apoptosis and viability is currently conflicting.
Round 2
Reviewer 1 Report
Authors addressed all my previous comments
Reviewer 2 Report
Authors responded to my comments, but no additional findings were provided in this revision process. Since the current findings are just preliminary data, this reviewer strongly hopes that authors further investigate the role of miR155 in ischemic heart disease.